# Modeling of Nonhydrostatic Dynamics and Hydrology of the Lombok Strait

**Alexey Androsov [1,2,\*], Naum Voltzinger [2], Ivan Kuznetsov [1] and Vera Fofonova [1]**

[1]  Helmholtz Centre for Polar and Marine Research, Alfred Wegener Institute, Am Handelshafen 12, 27570 Bremerhaven, Germany; ivan.kuznetsov@awi.de (I.K.); vera.fofonova@awi.de (V.F.)

[2]  Shirshov Institute of Oceanology RAS, 36 Nahimovskiy Pr., 117997 Moscow, Russia; lenna30@mail.ru

\*  Correspondence: alexey.androsov@awi.de

**Abstract:** The long-wave dynamics of the Lombok Strait, which is the most important link of the West Indonesian throughflow connecting the Pacific and Indian Ocean waters, was simulated and analyzed. A feature of the strait is its extremely complex relief, on which water transport creates a field of pronounced vertical velocities, which requires consideration of the nonhydrostatic component of pressure. The work presents a 3-D nonhydrostatic model in curvilinear coordinates, which is verified on a test problem. Particular attention is paid to the method of solving the 3-D elliptical solver for a nonhydrostatic problem in boundary-matched coordinates and a vertical σ level. The difference in transport through the Lombok Strait is determined by the difference in atmospheric pressure over the Pacific and Indian Oceans. Based on the results of the global simulation, the role of these factors in terms of their variability is analyzed, and the value of nonhydrostatic pressure in the dynamics of the Lombok Strait is revealed and evaluated. The vertical dynamics of the Lombok Strait are considered in detail based on hydrostatic and nonhydrostatic approaches.

**Keywords:** strait; numerical model; tidal dynamic; transport; hydrostatic/nonhydrostatic; energy; residual circulation; nonlinearity

---

## 1. Introduction

The Lombok Strait is the most important element in the system of passages of the Indonesian Archipelago. The Pacific waters, entering the Indonesian seas, form two branches: one, following to the east, finds an outlet to the Indian Ocean through the Banda Sea along the island of Timor; the second, western, lying between the islands of Kalimantan and Sulawesi, through the Makassar Strait and the Java Sea, goes into the Indian Ocean through the Lombok Strait (Figure 1a). The transport through the strait of several $Sv$ (1 $Sv$ = $10^6 \text{m}^3/\text{s}$) varies depending on the season, the variability of global characteristics of the boundary oceans and local characteristics of monsoon. The role of some other straits in the Western Indonesian throughflow (ITF) is rather insignificant.

The Lombok Strait, with its total length of around 60 km and width ~30 km, is located between the islands of Bali and Lombok (Figure 1b). An island of Nusa-Penida in the southern part separates the passage into two branches: a shallow western Bandung Branch along with the Bali and an eastern one, along with the Lombok Island, carrying three-quarters of inflow. The main feature of the strait morphometry is a sea mountain in its narrowest part. The depth above the sea mountain is around ~250 m; the depth increases over 30 km up to 2000 m southward and rapidly fits the oceanic bottom slope.

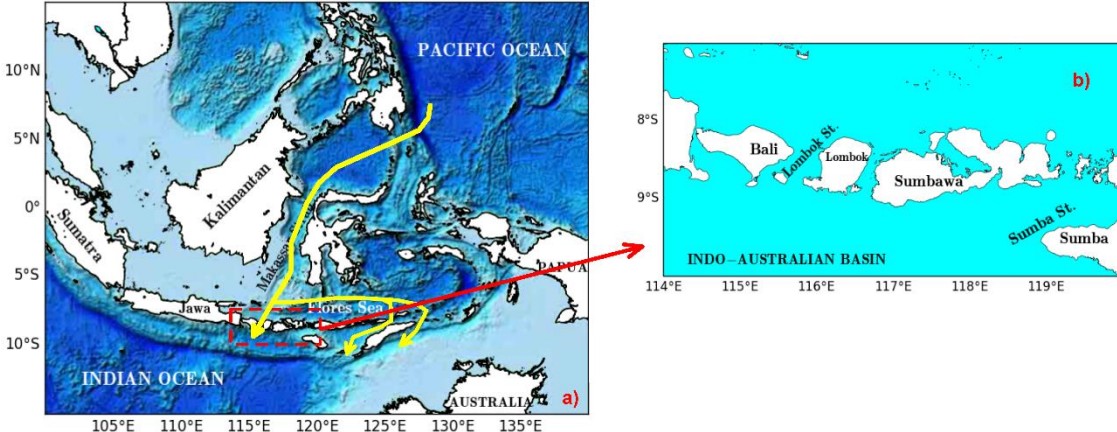

**Figure 1.** (**a**) The Indonesian archipelago; yellow arrow shows the water exchange between the Pacific and Indian Oceans. (**b**) The location of Lombok Strait in the Indonesian archipelago.

*Hydrophysical characteristics of the Lombok Strait.*

Water exchange through the Indonesian Straits is characterized by a generally stable transfer to the Indian Ocean. This can be judged by indices of the Pacific water masses transported in the Indian Ocean to the South of Africa and penetrating further into the Southern Atlantic [1].

Water transport through the straits of the ITF occurs due to the pressure gradient between the Pacific and Indian Oceans, generated by winds of tropical oceans [2]. The transport is variable [3,4] and depends on annual sea level variability in the Western Pacific. The mean value of level oscillations for twenty years is ~15 cm, with annual variability ~5 cm and maximum ±28 cm [5].

Transport through the Lombok mostly depends on the conditions at its northern boundary, at the same time being under the direct influence of strong tides of the Indian Ocean. According to Murray and Arief [6], the mean annual transport equals −1 *Sv* with $Q_{max} = -4$ *Sv* during southern monsoon. In Hautala et al. [7], the mean biennial water transport is determined as $Q = -2.6 \pm 0.8$ *Sv*; the authors of [4] report similar mean values over the observational period from 2004 to 2006 as $Q = -2.6$ *Sv*.

In tidal dynamics, over the sea mountain, a semi-diurnal wave dominates, with current values of around 3.5 m/s [8,9]. Nonlinear interaction of diurnal and semi-diurnal waves generates strong tide with a period of around 14 days [10].

Thermohaline currents in the Lombok Strait have a two-layer structure formed due to the difference in hydrodynamic characteristics of the ocean at its boundaries. The Pacific water inflow in the ITF has a higher temperature and lower salinity as compared to the waters of the Indian Ocean in the south of the strait. The interface is located at a depth of around 300 m. The influence of baroclinicity on tidal waves determines their variability, especially in the vicinity of the sea mountain, where a barotropic–baroclinic interaction is particularly intensive [11]. Interaction of semi-diurnal waves leads to strong vertical mixing of the Pacific waters and modification of their characteristics while passing through the Lombok Strait [12]. The Lombok is noteworthy in one interesting aspect going beyond oceanography. This deep strait with a steep sea mountain and a strong current above it turned out to be an absolute obstacle for the migration of some fauna species, the so-called Wallace line, due to which the exotic wildlife of Australia became isolated from the fauna of South-East Asia [13].

Database on the Lombok contains sporadic, fragmentary and irregular observations. The existing concepts are based on the information obtained from the research programs such as the program of International Acoustic Experiment, a TOPEX-POSEIDON program, particularly the part which considers the seas of the Indonesian archipelago and the satellite observations of surface indications of inner waves [8,14,15]. Another international program, INSTANT, which conducted measurements of currents in the Lombok was carried out in 2004–2006 to justify the project of installation of wave energy conversion turbines [16].

*Modeling of the Lombok Strait.*

The modeling of the Lombok Strait and the adjacent seas of the Indonesian archipelago is the focus of a number studies aimed at reproducing the main features of the internal and barotropic tidal dynamics of the region under various constraints in the formulation of the problem. In Visser [17], the authors analyzed the character of the instability of a current in a two-layer fluid above the Lombok sea mountain, leading to the generation of internal waves [18,19]. To simulate the internal waves in the Lombok, the authors of Nining et al. [20] used the Massachusetts Institute of Technology general circulation model (MITgcm). In Aiki et al. [21], internal waves and their energy were studied in an idealized domain with a 2-D model. Another direction is represented by models for studying the variability of the Indonesian Seas and their tidal dynamics: 2-D model [3], $1\frac{1}{2}$—layer model [22], 3-D hydrostatic version of ROMS (Regional Ocean Model System) [11] and also the 3-D hydrostatic model of the coastal zone of the ocean as applied to the Indonesian seas [23]. Assessment of water transport through the Lombok Strait was carried out with the global models. In Semtner and Chervin [24,25], the value of transport in the annual cycle as 15–18 *Sv* is obtained on a uniform grid of 0.5 degrees. Similar transport estimate is given also in Miyama et al. [3] based on a coarser mesh. The variability of the ITF and its connection to the fluctuations of the Pacific Ocean currents were modeled by England and Huang [26]. The high resolution global oceanic circulation model on the grid $0.1° \times 0.1°$ with 54 vertical levels in the domain from 75° S to 75° N was adopted to investigate the interseasonal variations in the region of the Eastern Indian Ocean and ITF straits [27].

Modeling the dynamics and hydrology of Lombok itself are virtually absent. The realistic modeling of tidal dynamics in the strait should take into account nonhydrostatic (*Nh*) effects, which are generally neglected in large-scale models, due to the high computational costs of calculating the *Nh* pressure corrections.

The presence of a sea mountain, as well as two-layer stratification, is typical for many straits due to the difference in water density at the boundaries between two basins [28–30]. A sea mountain determines the most essential features of straits, such as pronounced nonlinearity and presence of significant high-frequency modes in the spectra of main tidal waves; barotropic–baroclinic interaction and high variability of the vertical density profiles; generation of solitons at the sea-floor slopes and a sharp manifestation of *Nh* effects in the intervals of the tidal cycle extremes [31]. All of these factors are present in the Lombok dynamics, the modeling being also complicated by an exceedingly jagged bathymetry and high seasonal and interannual variability of the Lombok's hydrophysical fields.

The level of modern computational fluid dynamics is quite sufficient for the development of *Nh* modeling, and it is developing everywhere and quickly. This is evidenced by individual works and emerging system models with the capabilities of *Nh* applications [32–36]. Despite the variety of *Nh* models, they did not find wide application in practical modeling, and this is primarily due to the algorithmic exact solution and high computational cost.

Realistic simulations of the dynamics and hydrology of the Lombok Strait demand refusal from using the *Hs* approximation and construction of the *Nh* model. Such a model will correctly reproduce vertical velocities on mountain relief, determined by the dynamical component of pressure and horizontal components of the Coriolis acceleration.

Our aim is twofold. First, we present a model that solves the *Nh* primitive equations in a curvilinear coordinates system with the vertical *σ* coordinate and test it. Second, we apply it for the Lombok Strait—a very complex object that requires consideration of *Nh* effects. The main goal of the proposed study is to analyze the difference in the vertical structure of dynamic fields in *Hs* and *Nh* statements. For this region, we concentrate on tidal dynamics occurring on the background of regular transport between the Pacific and Indian oceans. This work, as far as we know, is the first initiative in this direction.

The next section presents a 3-D *Nh* boundary-value problem for the equations of dynamics and hydrology of the strait. A key element of the problem formulation is an assignment of the conditions at the northern boundary of the strait. The variation of such conditions is closely connected with irregular oscillations of tropical currents of the Pacific Ocean and, therefore, it would be pointless

to use fixed values of the boundary conditions. For this purpose, a preliminary procedure for the determination of normal annual difference of levels between the two oceans is carried out using the results from FESOM1.4 global simulations [37,38], which allows us also to estimate important characteristics such as the transverse level of distortion at the northern boundary. Section 3 considers some of the simulation results, i.e., water transport through the Lombok, dynamics of the summary tide, *Hs* versus *Nh* intercomparison in the maximum norm for the M$_2$ wave. Particular attention is paid to the analysis of the vertical structure of hydrodynamic characteristics and the analysis of nonlinearity fields in the Lombok Strait for *Hs* and *Nh* tasks. In Section 4, the conclusions are presented. In the two parts of the Appendix, we present a method for solving the *Nh* problem in a curvilinear coordinate system with $\sigma$ vertical levels, based on a combination of explicit–implicit schemes. Additionally, the verification and validation of the offered approach for *Nh* computation are presented.

## 2. Model

The theory of long-wave motions in conventionally based on the hydrostatic (*Hs*) approximation [39]. Its validity is related to the parameter $\varepsilon = H^2/L^2 \ll 1$, where $H$ is the characteristic depth, and $L$ is the characteristic length. Expanding the 3-D Euler equations in powers of $\varepsilon$, the first approximation leads to the *Hs* shallow water equations, and the second one contains a dynamical dispersion correction to the *Hs* pressure [40]. These statements are valid to the extent that the characteristic scales keep their inherent sense. In case of a sill, the characteristic scales lose their global meaning: the depth may be sharply varying, changing the wavelength of a traveling long wave. In these conditions, more general *Nh* equations can be required to simulate dynamics in the sill region.

We assume that an undisturbed water surface coincides with a horizontal *X0Y* plane of the right-hand Cartesian coordinate system, an *0Z*-axis being directed upwards. For domain $Q_T = Q \times [0 \le t \le \widetilde{T}]$ where $Q$ is a 3-D domain limited with a free surface $\zeta(x, y, t)$, bottom $z = -h(x, y)$, and a side surface $\partial Q$, $x, y \subset \Omega$, $-h \le z \le \zeta$; $0 \le t \le \widetilde{T}$, the equations of 3-D motion (1)–(2), continuity (3), temperature and salinity (4), state of sea water (5) and vertical average equation for free surface (6) are considered:

$$\frac{d\mathbf{v}}{dt} + g\nabla_2\zeta + g\rho_0^{-1}\nabla_2 \int_z^\zeta \rho'dz + \frac{1}{\rho_0}\nabla_2 q + f_v\mathbf{v}' = -f_h w + (\vartheta\mathbf{v}_z)_z + \nabla_2(K\nabla_2\mathbf{v}), \tag{1}$$

$$\frac{dw}{dt} + \frac{1}{\rho_0}q_z = f_h u + (\vartheta w_z)_z + \nabla_2(K\nabla_2 w), \tag{2}$$

$$\nabla \times \mathbf{u} = 0, \tag{3}$$

$$\frac{d}{dt}\Theta_i = \frac{\partial}{\partial z}\vartheta_{\Theta_i}\frac{\partial\Theta_i}{\partial z} + \nabla_2\left(K_{\Theta_i}\nabla_2\Theta_i\right), \tag{4}$$

$$\rho(x, y, z, t) = \rho(\mathrm{p}, \Theta_i), \tag{5}$$

$$\frac{\partial\zeta}{\partial t} + \nabla_2 \int_{-h}^\zeta \mathbf{v}dz = 0, \tag{6}$$

where $d/dt = \partial/\partial t + \mathbf{u} \times \nabla$, $\mathbf{u} = (u, v, w)$ and $\mathbf{v} = (u, v)$ are the velocity vectors; $\mathbf{v}' = (-v, u)$; $\zeta$ is the sea surface height; $\nabla = (\partial/\partial x, \partial/\partial y, \partial/\partial z)$ and $\nabla_2 = (\partial/\partial x, \partial/\partial y)$ are the gradient operators; $g$ is the gravitational acceleration; $\rho_0 = 1024.95$ kg/m$^3$ is the reference density; $f_h = 2\omega\cos\varphi$ is the horizontal component of the Coriolis parameter; $\omega$ is angular velocity of the Earth rotation; $\varphi$ is the latitude; the vertical component $f_v = 2\omega\sin\varphi$ is the vertical component of the Coriolis parameter (is relatively small in the equatorial zone); $\mathbf{v}' = (-v, u)$; $\mathbf{w}' = (w, 0)$; $\mathrm{p} = p_\Gamma + q$ is the sum of hydrostatic pressure $p_\Gamma$ and *Nh* pressure component—$q$; $\vartheta$, $K$ are the coefficients of vertical and horizontal turbulent mixing; $\Theta_i$ are the constituents of density: $i = 1, 2$; $\Theta_1 = T$—temperature, $\Theta_1 = S$—salinity; $\vartheta_{\Theta_i}, K_{\Theta_i}$ are the

coefficients of turbulent diffusion. The default scheme to compute the vertical viscosity and diffusivity in the system of Equations (1), (2) and (4) is based on the Prandtl–Kolmogorov hypothesis of incomplete similarity. According to this, the turbulent kinetic energy $b$, the coefficient of turbulent mixing $\vartheta$ and dissipation of turbulent energy $\varepsilon$ are connected as $\vartheta = l\sqrt{b}$, where $l$ is the scale of turbulence, $\vartheta_\Theta = c_p\vartheta$, $\varepsilon = 0.046b^2/\vartheta$ [41]. Prandtl's number $c_p$ is commonly chosen as 0.1 and sets the relationship between the coefficients of turbulent diffusion and viscosity. A detailed solution of turbulent closure equation is given in Androsov et al. [42].

The boundary value problem (1)–(6) in curvilinear coordinates (see Appendix A) is solved with the following parameters: $f_h = 1.46 \times 10^{-4}$ rad/s; bottom friction is $3.8 \times 10^{-3}$; $K = 50$ m$^2$/s; time step is 35 s. Three-dimensional temperature and salinity field are determined according to [8]. These data were digitalized and interpolated onto the computational mesh with a vertical resolution of 71 $\sigma$ layers. Bathymetry data have been compiled from data provided by TCARTA (https://www.tcarta.com/) with a resolution of 90 m by horizontal (Figure 2a). Sea level oscillations data in the Lombok Strait were obtained from the database of the ocean tides TPXO6.2 [43,44].

For assignment of sea level at the boundaries of the Lombok Strait, the interpolation of the results of FESOM1.4 global modeling of sea level on the unstructured grid is used [38]. The Figure 2b shows the annual average sea level at 9 points in the global model for the area of interest. Note that the simulation of the global model is carried out without tidal forcing.

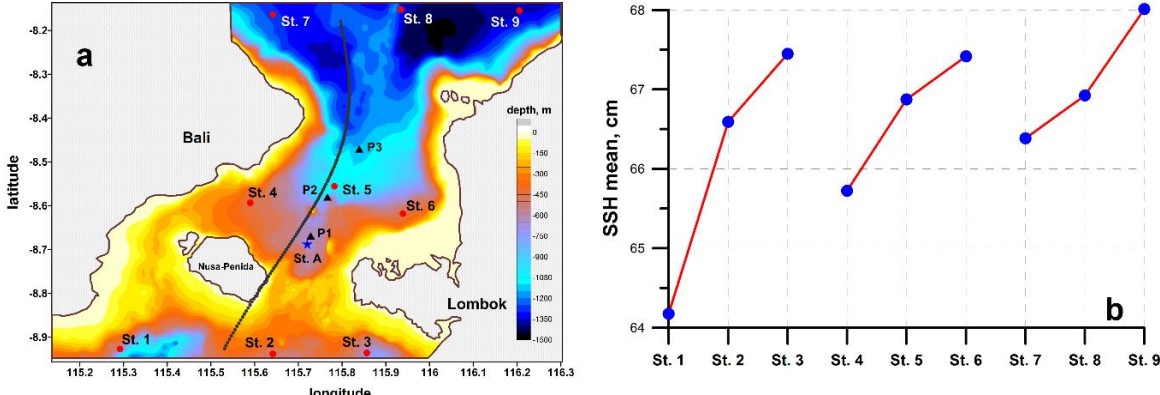

**Figure 2.** (**a**) Bathymetry map of the Lombok Strait with the location of the 9 stations for sea surface elevation analysis from FESOM1.4 global model [38] and black triangles (P.1–P.3) indicate the position of three points for spectral analysis; blue star indicates the position of station A; black line indicates cross-section along the strait. (**b**) The mean sea level for several years (2000–2009) at the chosen stations 1–9.

A curvilinear grid $\Omega_\Delta$ in the strait domain (see Figure 3), $\Omega$ is constructed using an elliptical method [45] with orthogonalization at the boundary $\partial\Omega$. The computations in the domain $Q_\Delta$ were performed on a grid $121 \times 121 \times 71$ with spatial resolution ranging from 315 to 1340 m. Vertically, we use 71 $\sigma$ level with a high resolution near the bottom and surface.

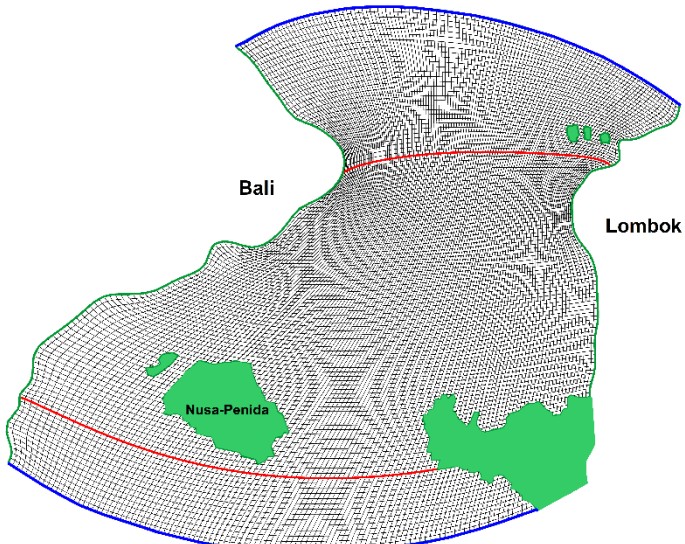

**Figure 3.** The computational grid $121 \times 121 \times 71$; $\Delta_{max}$ =1340 m, $\Delta_{min}$ = 315 m. reen line represents the solid boundary, blue line represents the open boundary and red line represents the domain with Nh pressure.

## 3. Results

Discussion of the simulation results will begin with the analysis of computations performed in the *Hs* formulation for $M_2$ wave and summary tide. The purpose of these computations is to select the optimal scenario for additional sea level values at the northern boundary of the modeled domain.

As it was already mentioned, a distinctive feature of the Lombok dynamics is its high variability in the fields of ocean currents and the influence of regional monsoons. Figure 4 gives an idea about sea level oscillations during the period of 2000–2009. At interannual variability of ~10 cm, the maximum amplitude of oscillations amounts to around 25 cm; a sawtooth character of interannual oscillations is the result of seasonal monsoons.

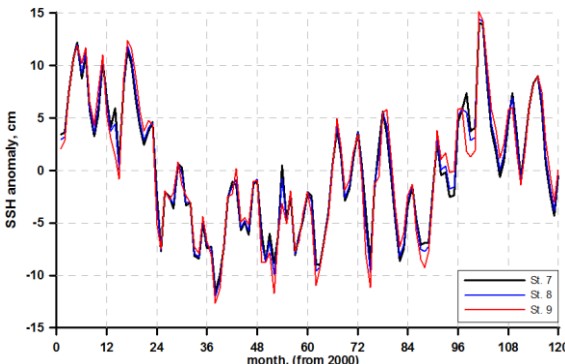

**Figure 4.** The sea surface elevation from the global model FESOM1.4 simulation [38] for Stations 7–9 shown in Figure 2.

### 3.1. Transport through the Lombok

The high variability of the water exchange rate through the Lombok makes it pointless to compute water transport at fixed boundary conditions. In this situation, it seems more appropriate to consider transport depending on the variation of boundary conditions within a range of real deviation from the mean interannual values of sea level at the boundary given by the global model. We performed three sensitive runs for different scenarios of levels at the open boundary. In Figure 5a, water transport is shown for the additional sea level values at the northern boundary: $Z_0 = 2.5 \div 0.5$ cm, $Z_1 = 5 \div 1$ cm,

$Z_2 = 10 \div 2$ cm, where the first number indicates the sea level at the eastern side of the north open boundary, and the second number denotes the sea level at the western side of the north boundary (see Figures 2b and 4). The maximum difference in a tidal cycle of $M_2$ wave between the cases $Z_0$ and $Z_1$ falls at the interval $3T/4$ ($T$ is tidal period of the $M_2$ wave) and totals around 1 $Sv$, while the difference between $Z_0$ and $Z_2$ in this interval is quite significant, reaching $\sim 4.8$ $Sv$. The result is obvious and interesting just quantitatively. The result presented in Figure 5b is more informative: here, the basic variant $Z_0$ is compared with the results obtained at two different conditions, i.e., when the sea level is specified without tide and when the only tide is specified at the northern boundary. In the first case, we have quasi-constant water transport $Q = -1.92$ $Sv$, while in the second case, we obtain mean transport $Q \approx +0.5$ $Sv$ (the negative sign corresponds to southern transport). Hence, it follows that strong tides of the Indian Ocean prevent the Pacific water transport.

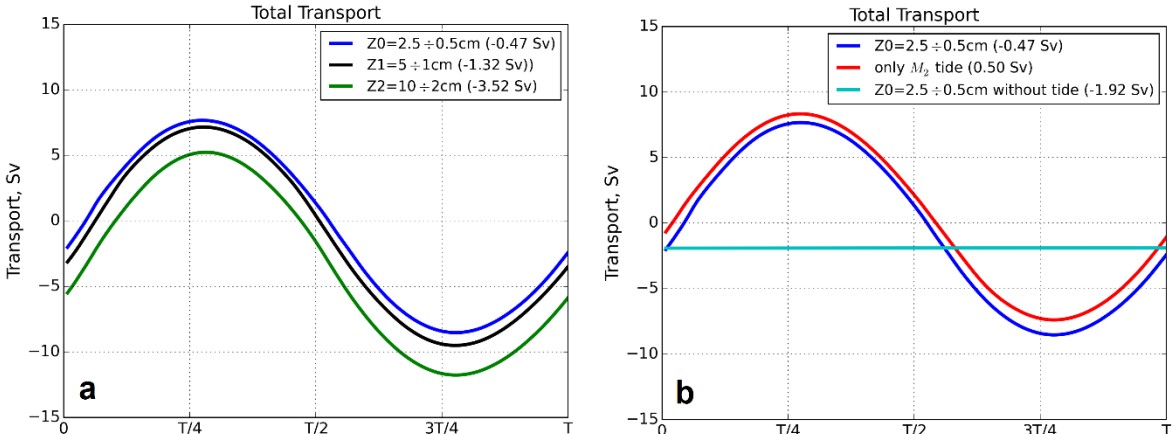

**Figure 5.** (**a**) transport through the strait under variants of boundary conditions on the northern boundary for the one tidal period of the $M_2$ wave (T = 12.42 h). (**b**) tide contribution to transport through the strait.

To make a representative comparison of the modeling results with the observation data, the computation of the summary tide for four semidiurnal and three diurnal harmonics $M_2$, $S_2$, $N_2$, $K_2$; $K_1$, $O_1$, $P_1$ with a period of 29.5 days was carried out. Maximum velocities in a summary tidal cycle are very close to a maximum mark of 3.5 m/s in Visser [17]. Transport estimates from various sources and various time intervals are presented in Table 1. Apparently, transport through the passage of the Lombok has strong variability, changing in a few $Sv$. Assessment of transport based on our modeling is close to the results in Hautala et al. [7].

**Table 1.** Transport across Lombok Strait. Negative values indicate flow toward the Indian Ocean.

| Source | Transport, $Sv$ | Comments |
|---|---|---|
| Chong et al. [46] | −0.2 | Mean surface transport in 1996 |
| Chong et al. [46] | −1.6 | Mean surface transport in 1997 |
| Hautala et al. [7] | −1.9 ± 0.3 | March 1997, 0–100 m |
| Hautala et al. [7] | −1.5 ± 0.3 | March 1998, 0–100 m |
| Feng et al. [47] | −2.6 | Mean water transport for 2004–2006 |
| Model results | −1.81 | Surface transport with a boundary condition of variant $Z_2$ at the northern boundary |

It is obvious that such a comparison can be qualitative only, firstly, due to the variability of transport and, secondly, because the model does not take into account the influence of seasonal monsoons.

Local values of the velocity in a surface layer with thickness of 100 m located in the middle part of the strait (not shown here) are in good agreement with the mean annual and mean monthly observation data [9,46]; however, without more detailed information, the above agreement should be considered as a coincidence.

The modeling of the summary tide allows us to quantify the contribution of the Indian Ocean tides to the transport through the Lombok Strait. It is found that the water transport based on just the $M_2$ wave forcing with a boundary condition $Z_1$ corresponds to the water transport of the summary tide with a condition $Z_2$. As it was already mentioned above, the Indian Ocean tides prevent southern transport, and the results obtained numerically prove this fact.

### 3.2. Estimation of the Influence of Nh Factor on the Strait's Hydrodynamics

The tidal dynamics simulated in the *Hs* approximation quite satisfactorily describe the average characteristics of dynamics and water exchange between oceans. Despite this, significant effects of *Nh* can be expected in the region of morphometric features and at times of extreme tidal cycle conditions [36]. In the dynamics of the Lombok Strait, with its complex underwater mountain bathymetry, the role of *Nh* should be quite pronounced over almost the entire area of the strait, especially in its northern part. To confirm and evaluate the above assumption, the Lombok Strait dynamics were modeled using *Nh* and *Hs* approaches. The difference in the solutions was determined in the northern and southern parts of the strait. Figure 6 presents such difference $\delta\mu$ for vector $\boldsymbol{\mu} = (\mathbf{v}, \zeta, \rho')$ in a C-maximum norm—$\| \delta\boldsymbol{\mu} \|_C = \max_N(\boldsymbol{\mu}_{Nh} - \boldsymbol{\mu}_{Hs})$—for all the points of a grid domain $N$ during a tidal cycle of an $M_2$ wave. The difference in the two solutions for dynamic characteristics behaves similarly, achieving maximum values in the northern subdomain of the strait, where $\| \delta\zeta \|_C \approx 3$ cm, $\| \delta u \|_C \approx 25$ cm/s, $\| \delta v \|_C \approx 15$ cm/s at the maximum fluctuations of elevation of 70 cm and the maximum horizontal velocity reaching 250 cm/s in the interval $\sim T/2$ during tidal current changes when the dynamic component of pressure grows. The difference in solutions with regard to the baroclinicity manifests itself in a different way, i.e., it increases strongly in the beginning and, at the end of a tidal cycle, up to the values $\| \delta\rho' \|_C \approx 0.5$ kg/m$^3$ in the southern part of the strait, being under the influence of denser waters of the Indian Ocean. The role of *Nh* in $L_2$ (average) norm is small.

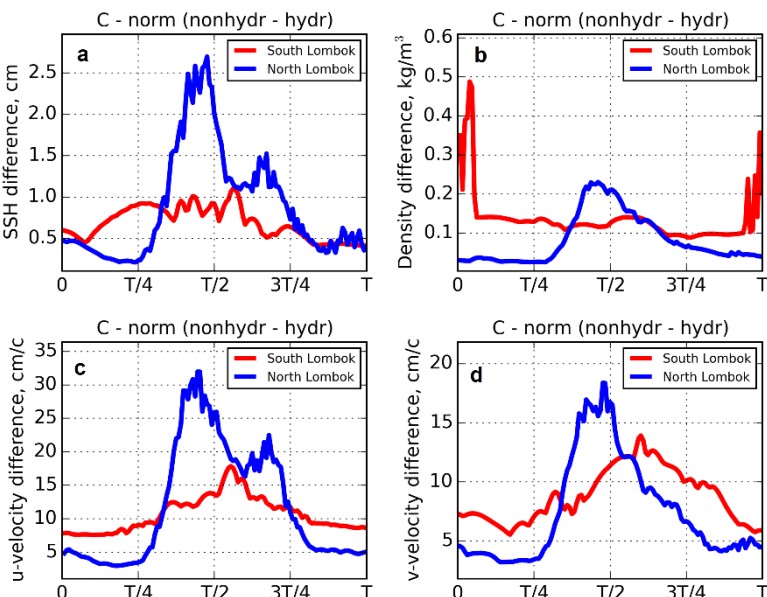

**Figure 6.** The difference in the solutions in *Nh* and *Hs* problems for the four components of the solution ((**a**) sea level, (**b**) density and (**c,d**) two components of horizontal velocity) in the C-norm.

As mentioned above, the consideration of *Nh* pressure is largely manifested in the vertical dynamics in the region of sea mountain and continental slopes. Figure 7 shows the results of the change in temperature and velocity in the cross-strait direction and vertical velocity at point A (Figure 2a) in depth over the tidal period of the $M_2$ wave in *Nh* (left panel of Figure 7), *Hs* (middle panel of Figure 7) and their difference (right panel of Figure 7). The difference in the temperature field can reach 0.1 °C

during the tidal cycle, and the spatial distribution of the difference is determined by a strong phase shift having a value of the order of one hour. At the same time, the maximum difference in temperature fields is reached at the moments of minimum energy (change in flow direction). The difference in velocity across the strait also has a phase shift, and the difference in values at this point can reach 12–15% of the absolute value.

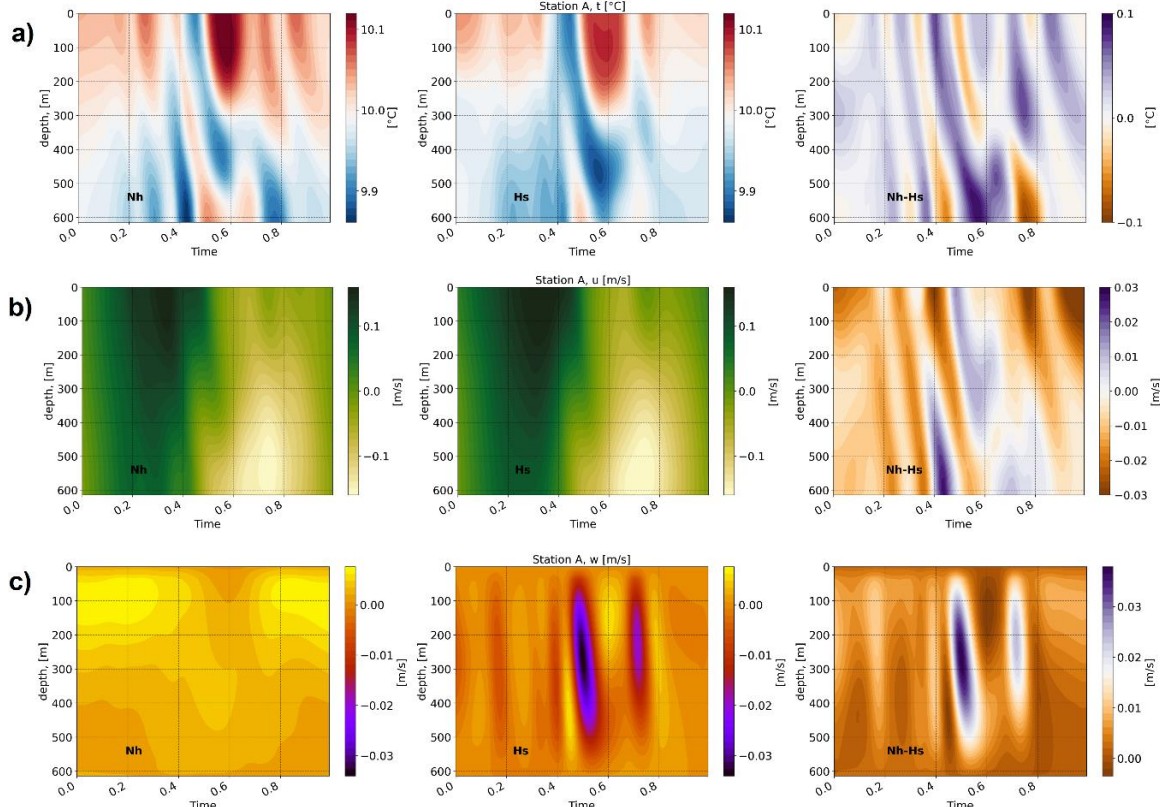

**Figure 7.** Hydrodynamic characteristics in point A (see Figure 2a) for the tidal period of the $M_2$ wave. (**a**) temperature; (**b**) cross-section velocity; (**c**) vertical velocity. Left column: *Nh* approach; middle column: *Hs* approach; right column: difference between *Nh* and *Hs*.

The role of *Nh* in long-wave motion is directly determined by the field of vertical velocities formed due to the dynamic component of pressure. Therefore, it is especially interesting to compare the vertical velocity values when solving the problems *Nh* and *Hs*. As can be seen from Figure 7c, the vertical velocity behavior in *Nh* and *Hs* has a completely different structure in time and space compared to the temperature fields and horizontal velocity. In the *Nh* approach, the vertical velocity has a pronounced periodic character, while in *Hs*, there is a noise structure in time and the periodicity of the tidal cycle is completely disturbed. The value of the vertical velocity *Hs* at some points in time exceeds *Nh* by an order of magnitude, which is especially evident at the moment of flow change.

Distribution of vertical velocity in the section along the axis of the strait in the *Nh* and *Hs* approaches is shown in Figure 8. Vertical sections are given at two points in time of the energy cycle of the $M_2$ wave—the first energy maximum and the second energy minimum. The *Hs* vertical velocity distribution along the axis of the strait during the tidal cycle has strong vertical instability in almost the entire area of the Lombok Strait. This is especially evident at the sharp bathymetry, where the strong change in currents in the vertical direction is clearly visible (a similar pattern of vertical currents can be observed in the test experiment given in Appendix B). The vertical velocity in the *Hs* approach can reach 10 cm/s in the area of the sea mountain.

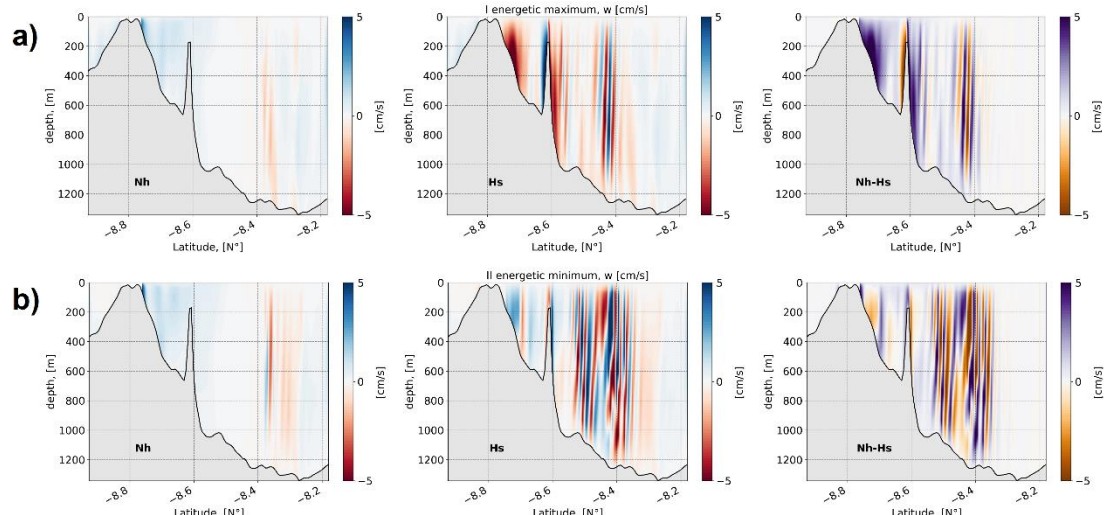

**Figure 8.** Distribution of vertical velocity in the section along the axis of the Lombok Strait (Figure 2a) in *Nh* and *Hs* approaches. (**a**) I energetic maximum; (**b**) II energetic minimum. Left: Nh approach; middle: *Hs* approach; right: difference between *Nh* and *Hs* vertical velocities.

The vertical velocity difference in the two approaches is shown in the right panel of Figure 8. It almost completely reproduces the unstable picture in *Hs* both in structure and in values. A similar spatial pattern is observed in the other phases of the $M_2$ tide.

The character of vertical instability in the case of *Hs* no longer has the periodicity of the $M_2$ wave but has a complex pronounced nonlinear character. The strong nonlinear processes that occur in this case will be analyzed in the next subsection.

### 3.3. Estimation of the Nh Factor Influence on Nonlinear Dynamic

Consideration of *Nh* pressure leads to significant spatial transformation of nonlinearity fields available in the *Hs* problem. Let us start by considering the results of the nonlinearity analysis of the sea surface elevation of the wave $M_2$. Figure 9a shows the spatial distribution of the sum of nonlinear terms' amplitudes ($M_4, M_6, \ldots, M_{18}$) in the *Hs* approach. As can be seen, the maximum nonlinearity falls on the shallow zone above the underwater ridge connecting the Nusa-Penida and Lombok islands. In this zone, the total amplitude of the nonlinear terms is 12 cm, which is approximately one quarter of the amplitude of the $M_2$ wave for this part of the domain. In the central part of the Lombok Strait, the amplitude of the total nonlinearity is also significant and reaches 2–3 cm (10% of the amplitude of the $M_2$ wave); the maximum values are achieved on sharp bathymetry (Figure 2a).

Figure 9b shows the difference in the sum of nonlinear terms of the sea surface elevation in the $M_2$ tidal wave spectrum between *Hs* and *Nh* tasks. The maximum difference reaches 2 cm in the area of the underwater ridge and has a wave character in spatial distribution. This kind of variability in the nonlinearity fields gives an indication of the difference in the residual circulation dynamic (not shown) in this area.

In the northern part of the strait, the difference in nonlinearity fields is due to the vertical velocity pattern, as will be shown below.

Figure 10 shows the fields of total nonlinear terms in the wave spectrum of the $M_2$ wave in the vertical velocity near the bottom. The choice of this vertical level is not accidental, since it is on the lower horizons at the end of the sharp slope where the maximum difference is observed in the density fields (see in Appendix B). Figure 10a,b shows the amplitudes of total nonlinear terms ($M_4, M_6, \ldots, M_{18}$) in *Hs* and in *Nh* approaches respectively. As can be seen, the spatial fields are significantly different from each other. In the *Hs* task, a strongly noisy character of nonlinearity is manifested in vertical velocity

at the sharp bathymetry in the northwestern and central part of the Lombok Strait. The absence of dispersion in the *Hs* approach for dynamics on the sharp slope leads to strong artificial nonlinearity.

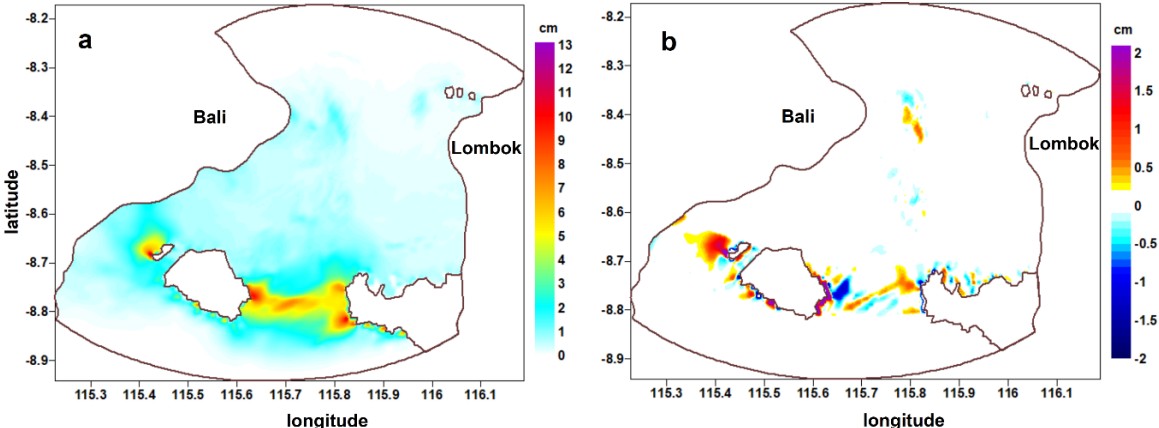

**Figure 9.** Nonlinearity in the sea surface height. (**a**) *Hs* approach; (**b**) the difference in the nonlinearity in the sea surface height between *Hs* and *Nh* approach.

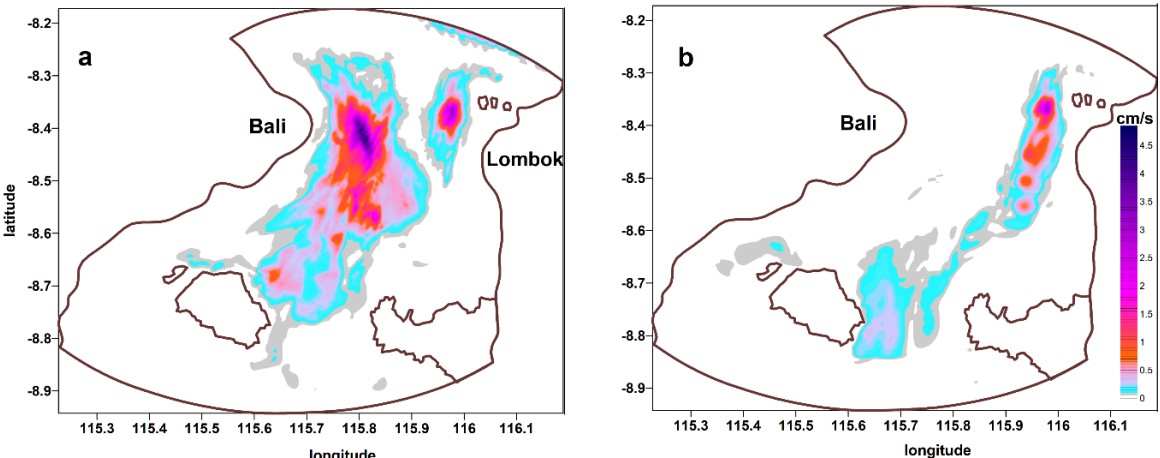

**Figure 10.** Nonlinearity in the vertical velocity near the bottom. (**a**) *Hs* approach; (**b**) *Nh* approach.

The amplitudes of the nonlinear vertical velocity members reach 3–4 cm/s and can exceed the amplitude of the tidal harmonic of the $M_2$ wave.

One can also note the presence of strong nonlinearity near the northern open boundary in the *Hs* approach. The absence of nonlinearity near open boundaries (the case of *Nh*) may indicate the correct matching of the barotropic and baroclinic signals near open boundaries.

Spectral analysis of vertical velocity is carried out for three points of the Lombok Strait. Figure 11 (left panel) shows the evolution of velocities in the *Hs* and *Nh* approaches for the tidal period of the $M_2$ wave. In the right panel, Figure 11 shows corresponding vertical velocity spectra.

The analysis of the vertical velocity in the *Hs* approximation gives a saturated spectrum of significant amplitudes practically over the entire frequency range at the points under consideration, which corresponds to the evolution of sawtooth waves in the $M_2$ cycle. The *Nh* spectrum at these points shows at most only two harmonics, the main $M_2$ and the first nonlinear. Moreover, their amplitudes are much lower than in *Hs*. This result is consistent with the nature of the evolution of the vertical velocity, which is here more variable for the *Hs* problem than for the *Nh* problem.

The results of the experiment allow us to conclude that the computation of vertical velocity in the *Hs* approximation not only can significantly change its values but also can also distort the

spectrum because hydrostatic vertical velocity is found from the continuity equation, which leads to the non-physical excessive nonlinearity.

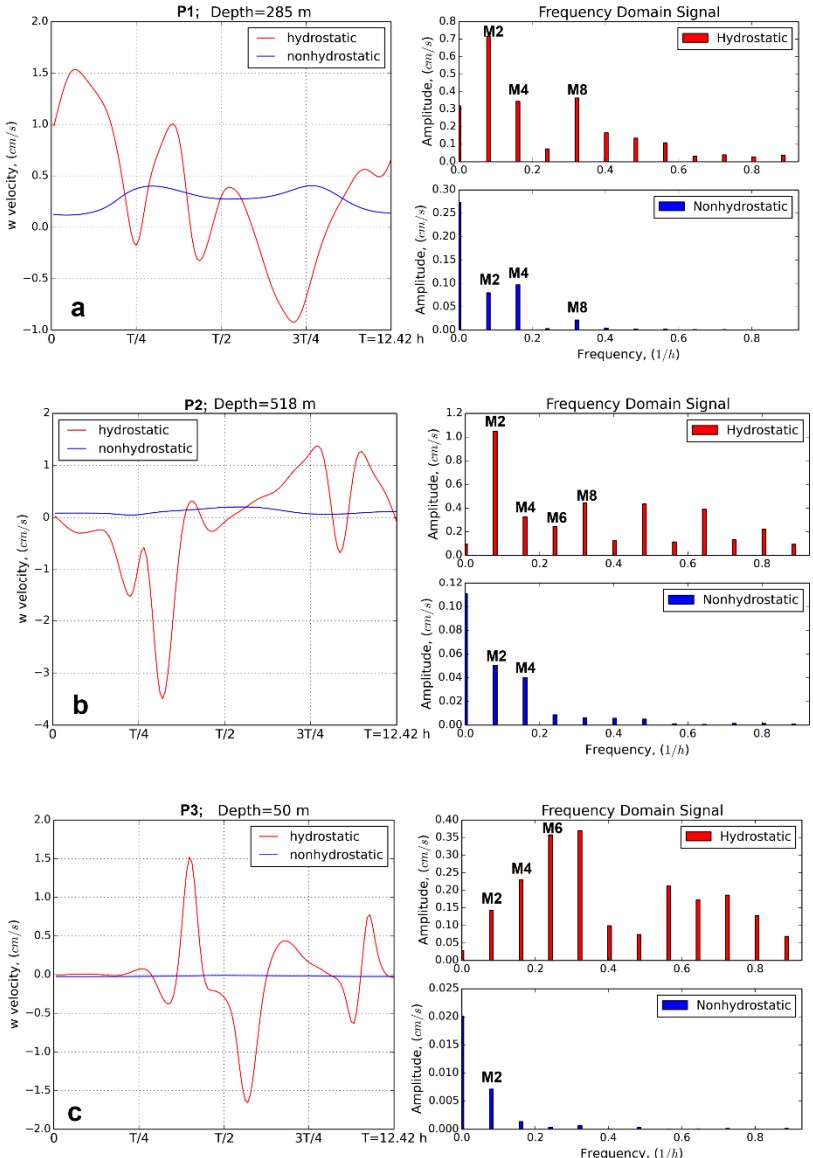

**Figure 11.** Left panel: comparison of the vertical velocity in the tidal cycle $M_2$ for *Nh* and *Hs* simulations; right panel: the spectra of vertical velocity at the points (**a**) P1; (**b**) P2; (**c**) P3 shown on the Figure 2a.

## 4. Conclusions

This paper presents model results of the dynamics in the Strait of Lombok, which is the most important component of the western branch of the Indonesian throughflow, on the basis of a 3-D regional model.

The aim of this work was twofold. The first was the modeling of the main features of the Lombok tidal dynamics against the background of the water transport between two oceans. Water exchange is imposed as the pressure gradient between the oceans by the northern boundary condition. Due to the limited database and high variability of dynamic characteristics of the strait caused by the currents' fluctuations in the Indian and Pacific Oceans, as well as by the unsteady influence of seasonal monsoons, the computations were carried out for several boundary conditions in the range of their mean multiannual variations determined on the basis of a detailed global model of the world ocean sea level oscillations. All of the obtained results contain the data on water exchange through the Lombok

Strait at variable realistic boundary conditions, as well as estimation of the role of the M$_2$ wave and the summary tide consisting of seven harmonics on the water exchange; comparison of the modeling results with the observation data took into account the summary tide. The second aim of this work was estimation of the role of *Nh* dynamics in the strait.

The most interesting of all of the results obtained is a quantitative estimation of the role of each of the two factors of water exchange through the Lombok Strait, i.e., the tide and pressure gradient between the oceans at the conditions when southern water transport normally occurs. When passing through the seas of the Western Indonesian throughflow, the Pacific tide energy is weakened and, in the absence of pressure difference between the oceans due to the direct influence of the Indian Ocean tides, the northern transport becomes dominant. The results of the model show that when only the M$_2$ wave is taken into account, the northern transport equals +0.4 *Sv*; under the influence of the summary tide, the northern water transport grows, so, to invert it and obtain a realistic southern transport, the pressure gradient is required in the form of a boundary condition with a level in excess of ~5 cm and transverse inclination ~2 cm.

Another important result concerns the role of *Nh* in the Lombok Strait dynamics. It is normally assumed that in a long-wave motion, a dynamic component of pressure can be neglected. Actually, this is not totally true in the Lombok Strait. The results presented in C-norm (maximum norm) and in L$_2$-norm (Euclidian norm) will be different. Smallness values in L$_2$-norm attest that, globally, *Hs* approximation is admissible, but the large value of the C-norm indicates the importance of taking into account *Nh* locally.

One such indicator of the need for accounting of *Nh* pressure is vertical velocity. The complex nonlinear behavior of *Hs* vertical velocity is a reflection of the absence of dispersion for this type of task.

We also note the complex nonlinear structure of hydrodynamic fields arising in the *Hs* approach. This nonlinearity is especially pronounced in vertical velocity fields and sea surface elevation.

Underlining this, the *Hs* and *Nh* approaches provide the same estimation for the water transport within different conditions. However, the *Nh* approach ensures the correct water mass transformation in the Lombok Strait. The artificially large vertical velocities and pronounced nonlinearity with its relative noise character in the straight in the frame of the *Hs* approach lead to extensive mixing in the channel and possible distortion of the outflow properties. This, in turn, can cause the wrong residual circulation pattern on the continental slope and coastal zones, wherein the *Hs* approach shows strong artificial nonlinearity.

In addition, we note an improvement in the consistency of the barotropic and baroclinic signal near open boundaries in the case of taking into account *Nh* pressure.

For areas such as Lombok, when transport changes are completely dependent on the variability of the boundary conditions, the modeling is performed by setting some average climatic characteristics at the boundary. This makes it possible to obtain only qualitative assessments of the regime. At the same time, the introduction of satellite observations allows us to move forward and move to "operational modeling", which uses the assimilation of the necessary information at the moment, as is done for weather forecasting.

This aspect of modeling, which combines actualization and forecasting, can be very useful to better understand hydrodynamic processes and to ensure good economic performance in the region.

**Author Contributions:** A.A., and N.V. designed experiments. A.A. set up and carried out the experiments. A.A., I.K. and V.F. analyzed and visualized the model as well as the observed data. A.A. and N.V. wrote the paper. A.A. and N.V. developed the GNOM model [48]. A.A., N.V., I.K. and V.F. contributed with discussions of the results. All authors discussed the results and commented on the paper at all stages. All authors have read and agreed to the published version of the manuscript.

**Funding:** This research was partly funded by the state assignment of FASO Russia (theme 0149-2019-0015). We acknowledge support by the Open Access Publication Funds of Alfred-Wegener-InstitutHelmholtz-Zentrum für Polar-und Meeresforschung.

**Acknowledgments:** The author wishes to acknowledge the valuable comments on this article from Sergey Danilov.

**Conflicts of Interest:** The authors declare no conflict of interest.

## Appendix A. Solution Method

We introduce a $\sigma$-coordinate in the vertical and horizontal curvilinear coordinates fitted to the shape of domain $\Omega(x, y)$. Let us consider the following transformation:

$$\xi = \xi(x,y), \ \eta = \eta(x,y), \ \sigma = H^{-1}(\zeta - z), \ t' = t, \tag{A1}$$

with the full depth $H = h + \zeta$, $-1 \le \sigma \le 0$ and the Jacobian $J = J_*H$, where $J_* = \frac{\partial(x,y)}{\partial(\xi,\eta)}$ is the plane Jacobian such that $0 \le J_* < \infty$. With an appropriate choice of four pairwise-opposite segments of a side surface $\Omega$, the domain $Q$ is mapped into a parallelepiped $Q^*$.

Equations (1)–(4) and (6) in the coordinates (A1) take the following form:

$$\frac{\partial}{\partial t}\mathbf{v} + \frac{\partial}{\partial \xi^i}\left(g\zeta + g\rho_0^{-1}H\int_\sigma^0 \rho' d\sigma + \frac{1}{\rho_0}q\right)\mathbf{e}^i + f_v\mathbf{v}' = -\mathcal{A}\mathbf{v} - f_\Gamma\mathbf{w}' + H^{-2}(\vartheta\mathbf{v}_\sigma)_\sigma + D\mathbf{v} \tag{A2}$$

$$\frac{\partial}{\partial t}\mathbf{w} + \frac{1}{\rho_0 H}q_\sigma = -\mathcal{A}\mathbf{w} + f_\Gamma u + H^{-2}(\vartheta\mathbf{w}_\sigma)_\sigma + D\mathbf{w}, \tag{A3}$$

$$\left(JU^i\right)_{\xi^i} + \left(J\widetilde{W}\right)_\sigma = 0. \tag{A4}$$

$$\frac{\partial}{\partial t}\Theta_i + \mathcal{A}\Theta_i = H^{-2}(\vartheta\Theta_{i_\sigma})_\sigma + D\Theta_i \tag{A5}$$

$$J_*\zeta_t + \left(J\overline{U}^i\right)_{\xi^i} = 0, \tag{A6}$$

Here, $\mathbf{e}^i = \nabla_2\xi^i$ are the contravariant basis vectors, $i = 1, 2$; the second term in (A2) represents the sum of *Hs* and *Nh* pressure gradients; $\mathcal{A} = U^i\partial/\partial\xi^i + W\partial/\partial\sigma$ is the advection operator; $U^i = \mathbf{v}\cdot\mathbf{e}^i$ is the contravariant horizontal velocities: $U^1 = U, U^2 = V$; $\overline{U}^i$ is the average on vertical contravariant horizontal velocities; $\xi^1 = \xi, \xi^2 = \eta$; $W = \sigma_t + \mathbf{u}\nabla\sigma$ is a contravariant vertical velocity; $D$ is an operator of the horizontal turbulent mixing [48]; $\widetilde{W} = W - \sigma_t$ [49].

Equations (A2)–(A6) are solved in the domain $\Omega^*(\xi, \eta)$. At solid boundary $U|_{\partial\Omega_1^*} = 0$; at the open boundary for outflow, $V$ is extrapolated; for inflow, both of the velocity components as well as the condition are assigned [48,50].

The initial conditions for Equations (A2)–(A6) are a divergence-free velocity vector $\mathbf{u}|_{t=0} = \mathbf{u}^0$ and constituents $\Theta_i|_{t=0}$.

The boundary value problem is solved at each time step $(k \to [k+1])\tau, k = 0, 1, \ldots, \hat{k} = [\hat{T}/\tau]$ in several stages. At the preliminary three stages, the computation of advection is fulfilled using the Strang scheme [51] and the *Hs* gradient of pressure $\rho_0^{-1}\nabla_2 p_h$ as a sum of its barotropic and baroclinic gradients. Leaving for simplicity the differential form for the difference operators in the Cartesian representation, we write a predictor equation at the fourth intermediate stage $\bar{k}$ as follows:

$$\begin{aligned} \frac{\left(\mathbf{v}^* - \mathbf{v}^{\bar{k}}\right)}{\tau} + \frac{1}{\rho_0}\nabla_2 p_h^{\bar{k}} &= \gamma_\mathbf{v}^{\bar{k}} \\ \frac{\left(w^* - w^{\bar{k}}\right)}{\tau} &= \gamma_w^{\bar{k}} \end{aligned}, \tag{A7}$$

where $\gamma_\mathbf{v}, \gamma_w$ contain the terms of advection, turbulent mixing and Coriolis acceleration. We represent the difference approximation of Equations (A2) and (A3) in the following form [48]:

$$\begin{aligned} \frac{\left(\mathbf{v}^{k+1} - \mathbf{v}^{\bar{k}}\right)}{\tau} + \frac{1}{\rho_0}\nabla_2\left(p_h^{\bar{k}} + q^{k+1}\right) &= \gamma_\mathbf{v}^{\bar{k}} \\ \frac{\left(w^{k+1} - w^{\bar{k}}\right)}{\tau} + \frac{1}{\rho_0}q_\sigma^{k+1} &= 0 \end{aligned}. \tag{A8}$$

Subtracting (A7) from (A8), we obtain an equation to determine *Nh* pressure:

$$\frac{\left(\mathbf{u}^{k+1} - \mathbf{u}^*\right)}{\tau} + \frac{1}{\rho_0}\nabla q^{k+1} = 0, \tag{A9}$$

which in the curvilinear coordinates with regard for (A1) has the following form:

$$\left(Jg^{ij}q_{\xi i}^{k+1}\right)_{\xi k} = \frac{\rho_0}{\tau}\left[(JU^*)_\xi + (JV^*)_\eta + \left(J\widetilde{W}^*\right)_\sigma\right] \tag{A10}$$

Oceanological modeling of mesoscale *Nh* is based on the most popular projection method for solving the Navier–Stokes equations [52,53]. The common difficulty in solving such a problem consists in the fact that the velocity field must satisfy, with very high accuracy, the continuity equation. The intermediate *Hs* velocity field can be perturbed by abrupt changes in topography, large density gradients and other hydrophysical factors, which is typical for the *Hs* fields. For the system of primitive equations, this is not a problem, as the continuity equation is enforced there, and for the Navier–Stokes equations, the perturbations of the intermediate *Hs* field are critical and the solutions may not converge. Oceanological models with a projection module use a wide range of methods for their numerical implementation with various forms of approximations [33,36,39,49,54].

The problem of high computational costs can be solved using the approach when the solution of the *Nh* problem is not sought in the whole computational domain but only in subdomains and over time intervals where one expects that the *Nh* pressure correction is substantial because of geometrical considerations [55,56]. The boundaries of the *Nh* zone should be located at some distance from the zones of violation of *Hs* conditions. In this case, the *Nh* pressure at these boundaries (see Figure 3) can be set to zero and the solution in subdomains is realized by a seamless and continuous computation that does not demand a procedure of matching the two problem solutions.

The *Nh* pressure is determined at each time step by solving the boundary value problem (A7)–(A9). Then, using dynamic pressure gradients, the *Nh* velocity vector $U^{k+1} = \left(U, V, \widetilde{W}\right)^{k+1}$ is found from Equation (A9), transformed to the boundary-fitted coordinates and, finally, the Cartesian velocity $\mathbf{u}^{k+1}$ is obtained from the following relations: $u = J_*\left(U\eta_y - V\xi_y\right)$, $v = J_*(V\xi_x - U\eta_x)$, $w = H\left(\widetilde{W} - u\sigma_x - v\sigma_y\right)$.

The Poisson equation for the Beltrami–Laplace operator (A10) is solved by a combination of the successive over-relaxation method (SOR) in planes $(\xi, \eta)$ and sweep method (three-point Thomas scheme) on $\sigma$. Let us write Equation (A10) in the following form:

$$(\mathcal{L} + L)q^{k+1} = \varphi^{*,k} + \psi^*, \tag{A11}$$

where

$$\mathcal{L} = \frac{\partial}{\partial\xi}\left(\alpha\frac{\partial}{\partial\xi}\right) + \frac{\partial}{\partial\eta}\left(\beta\frac{\partial}{\partial\eta}\right), \quad L = L_v = \frac{\partial}{\partial\sigma}\left(Jg^{33}\frac{\partial}{\partial\sigma}\right);$$

$$\varphi^{*,k} = \frac{\rho_0}{\tau}\left[(JU^*)_\xi + (JV^*)_\eta\right] - \left(Jg^{ij}_{\xi i}\right)_{\xi j}, \, i \neq j, \quad \psi^* = \frac{\rho_0}{\tau}\left(J\widetilde{W}\right)_\sigma; \quad \alpha = Jg^{11}, \, \beta = Jg^{22},$$

where $g^{ik} = \mathbf{e}^i\mathbf{e}^k$ are the components of contravariant metric tensor.

In grid domain $Q_\Delta^* = \{\xi_m, \eta_n, \sigma_s\}$, $m = 1, 2, \ldots, M$; $n = 1, 2, \ldots, N$; $s = 1, 2, \ldots, \bar{s}$, at each time step, the two problems are solved:

$$\begin{array}{c} \mathcal{L}_\Delta q_{m,n} = \varphi_{m,n}, \quad \forall_s \\ \text{with boundary conditions}: \quad \partial q/\partial\hat{n}\big|_{\partial Q_1^*} = 0, \quad \partial q/\partial\hat{n}\big|_{\partial Q_2^*} = \left(V^* - V^k\right)\rho_0/\tau g^{22} \end{array} \tag{A12}$$

and

$$\begin{array}{c} L_\Delta q_s = \psi_s, \quad \forall_{m,n} \in \Omega_\Delta \\ \text{with boundary conditions}: \partial q/\partial\hat{n}\big|_{\sigma=-1} = 0, \quad q\big|_{\sigma=0} = 0 \end{array} \tag{A13}$$

($\hat{n}$—outward normal).

Denoting $\gamma = 2(\alpha + \beta)$, we have the five-point approximation of (A12):

$$\mathcal{L}_\Delta q_{m,n} = \alpha_{m-1/2,n} q_{m-1,n} + \alpha_{m+1/2,n} q_{m+1,n} + \beta_{m,n-1/2} q_{m,n-1} + \beta_{m,n+1/2} q_{m,n+1} - \gamma_{m,n} q_{m,n} = \varphi_{m,n}. \quad (A14)$$

The SOR algorithm:

$$q_{m,n}^{v+1} = q_{m,n}^{v} + \frac{\omega}{\gamma_{m,n}} r_{m,n}^{v} \quad (A15)$$

where $v$ is the number of iterations, $\omega$ is the relaxation parameter, $1 \le \omega \le 2$. The best approximation for $\omega$ is as follows:

$$\omega = \frac{2}{1 + (1 - \varrho^2)^{1/2}},$$

where $\varrho$ is the spectral radius of matrix coefficients (A14). For the Poisson equation (A12) in square $M = N$,

$$\varrho = cos\pi/N \approx 1 - \frac{1}{2} \frac{\pi^2}{N^2}, \quad \omega \approx 2(1 - \pi/N)$$

and a factor of asymptotic convergence diminishing residual $\lambda = \omega - 1 = 1 - 2\pi/N$ [57]. The number of iterations $\overline{v}$ asymptotically necessary for decreasing residual up to $10^l$ is calculated as follows:

$$\lambda^{\overline{V}} = (1 - 2\pi/N)^{\overline{V}} = 10^l,$$

where $\overline{v} = \frac{ln10}{2\pi} Nl \approx \frac{Nl}{3}$.

Number of operations on one grid node in the SOR method are $N_0 = 11$ and for reducing the residual by $10^6$ times on a grid containing $O(N^2)$ points, several hundred iterations are needed. In our case, $N = M = 121, \overline{s} = 71$ and the number of operators which are necessary to obtain such accuracy at each time step is

$$R_{SOR} = N_0 \frac{Nl}{3} (N^2 \overline{s}) \approx 2.5 \times 10^9.$$

The effective three-diagonal Thomas algorithm when vectors of the sweep algorithm are calculated in advance demands on node only five operations and the total number of operations only $5 \times 10^7$. The structure of the algorithm for the implementation of the boundary-value problem for Equation (A12) is made up of several cycles, each of which consists of the SOR algorithm with reduced accuracy and implicit implementation (A13) in the vertical direction (sweep algorithm). For acceleration of convergence on each iterative step, the change in the directions of integration in all directions is used.

**Appendix B. Verification and Validation of the *Nh* Approach. Lock-Exchange on the Shelf Break**

Validation and verification of the *Nh* model solver are performed to show the feasibility and workability of the proposed approach. For this, a fairly widely used experiment was chosen, showing the importance of taking into account the *Nh* pressure in the interaction of two water masses on the continental slope [58]. In addition to comparing with published results [58], we also present vertical velocity fields reflecting errors associated with incorrect pressure accounting.

This test case is identical to the problem proposed by Haidvogel and Beckmann [59] and after by Heggelund et al. [58]. The area configuration, the vertical structure of a grid with 70 uneven layers having condensation at the bottom, initialization of the initial field of density was borrowed from Heggelund et al. [58]. The area had two open borders on which the free outflow area was set (Figure A1). The task was considered for 10 h from a condition of rest in two statements—*Hs* and *Nh*. Decisions were compared at a time point of 10 h when heavy water plume reached the end of the continental slope.

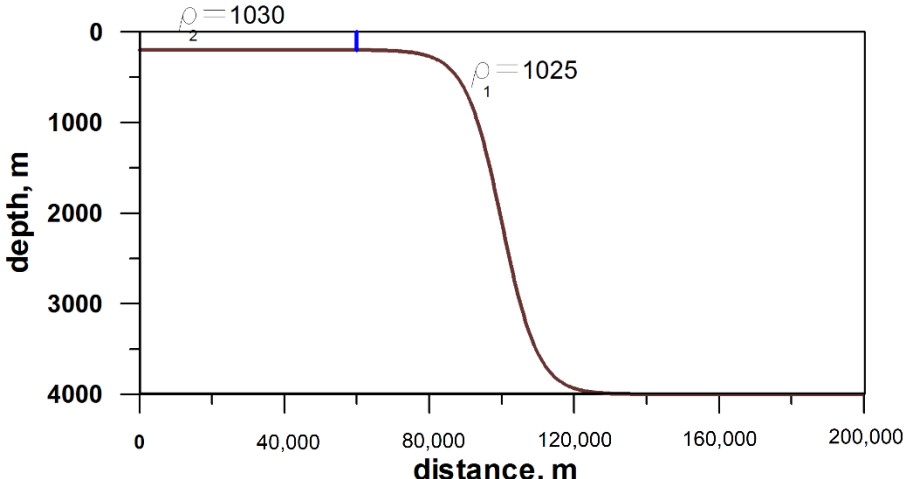

**Figure A1.** The initial setup for lock-exchange on shelf break.

In Figure A2, fields of density variation in two statements at the final time point of the experiment are presented. It is visible that in the *Nh* statement plume, it is more localized in space, and its front has a smooth characteristic. The *Hs* statement strongly has a high edge gradient of heavy water, and behind the front, the behavior of the plume has an oscillation characteristic. On a continental slope, the strong residual instability created by the movement of a plume is found.

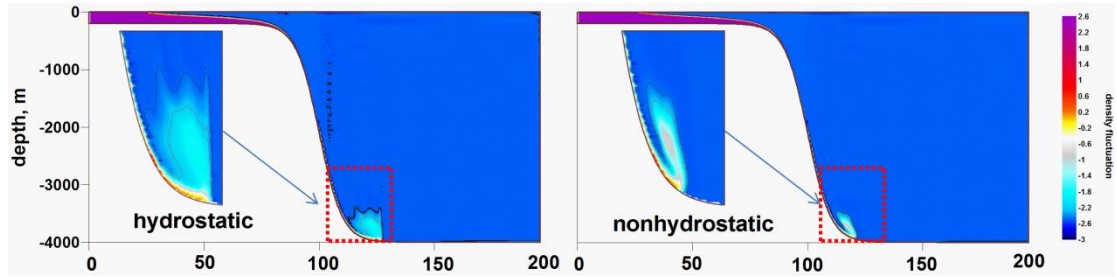

**Figure A2.** Density fluctuation distribution (kg/m³) for the hydrostatic approach (left) and nonhydrostatic (right) at time = 10 h.

The field of vertical velocities (Figure A3) in the *Nh* approach has a pronounced two-cell character. At the wavefront, there is a rise in water caused by the movement of the plume front and the lowering of the water masses behind the front. At the same time, in the *Hs* model, practically along the entire length of the continental slope, we observe a pronounced oscillatory characteristic in the field of vertical velocities. This structure of vertical currents is very similar in the Lombok Strait over the top of the steep sea mountain that is expressed in the sawtooth spectrum (see Figure 11).

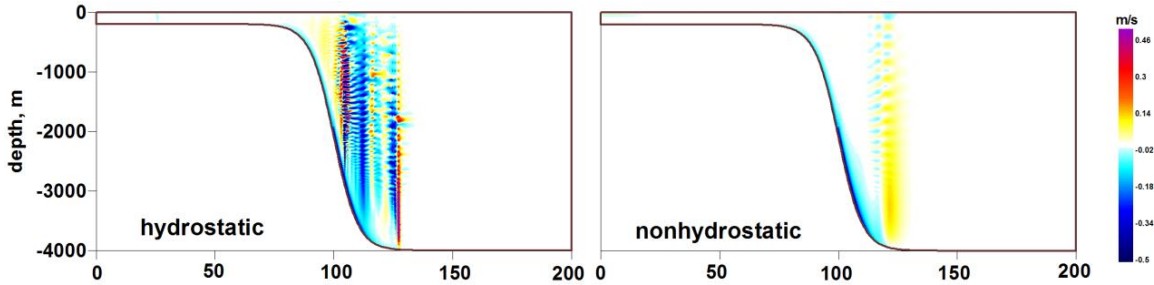

**Figure A3.** Vertical velocity distribution (m/s) for hydrostatic approach (left) and nonhydrostatic (right) at time = 10 h.

The most important benefit of accounting for the *Nh* pressure is that the density gradient of the plume is better preserved in this approach. This statement is well illustrated in Figure A4 (left panel), in which the error of a maximum and a minimum of density variation for all periods of simulation is shown. It is visible that in the *Hs* approach, oscillation of density behind the front of the wave (density deviations from a minimum variation) is strongly unstable at the time of achievement of the plume of the end of the continental slope. It is also clearly visible that at the initial moment when the mixing of two water masses starts, there is strong oscillation at the wavefront (density deviations from a maximum variation). In the right panel of Figure A4, the total energy of the two simulations is presented. It is shown that the *Hs* statement also leads to strong computing instability.

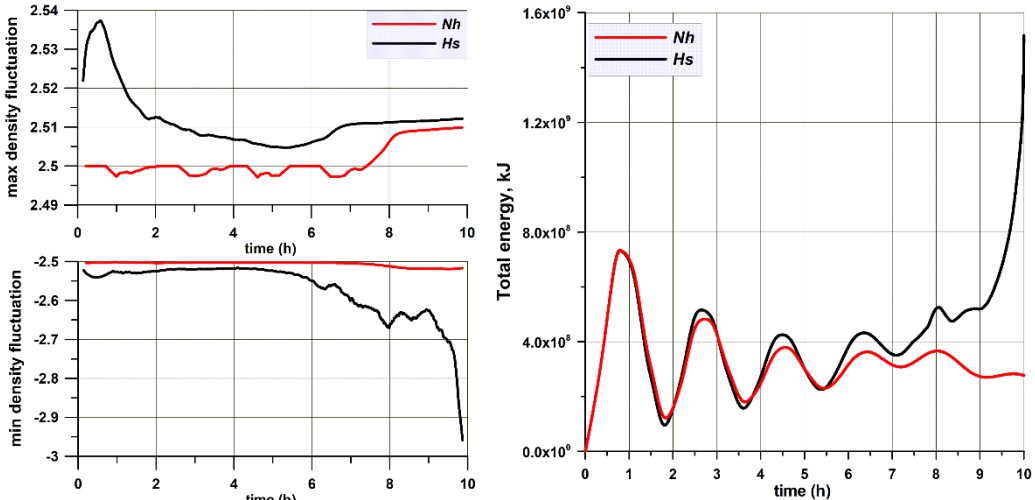

**Figure A4.** Left panel: maximum (upper) and minimum (bottom) variation of the density gradient for two approaches; right panel: total energy (kJ) for hydrostatic and nonhydrostatic statement for all periods of simulation.

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
