# Peer review of "Modeling of Nonhydrostatic Dynamics and Hydrology of the Lombok Strait"

_water, doi:10.3390/w12113092_

Round 1

Reviewer 1 Report

A very interesting problem, possibly more frequently reported in oceanology and marine journals. I have not come across such topics myself. An important and interesting issue, correctly presented.The modeling conclusions presented, even if for a limited database and estimates, are very valuable information. They allow you to make adjustments and further improve modeling. They contribute to a better understanding of the of water-exchange through the Lombok Strait.

Author Response

Grateful to the reviewer for the high appreciation of the work done.

Reviewer 2 Report

water-956746-peer-review-v1

L1-3 Title

The Title reflects the paper’s content accurately.

L10-24 Abstract

The abstract is determines the paper’s content and objectives in a very manifest and complete fashion.

 L26-203 1. Introduction

Quite comprehensive.

L205-314 2. Model

In L 206-216 there is no explanation of the derivation of the model equations which should be added. Present evidence regarding the negligibility of the geostrophic component. 

In the case of the Lombok strait there are three phenomena which may impact on the model considered.

  1. The KdV equation as seen in (Dwi Susanto, Mitnik, and Zheng 2005) based on (Osborne and Burch 1980) and (Syamsudin et al. 2019) specifically for the Lombok Strait or in (Grimshaw, Pelinovsky, and Talipova 1997).
  2. The role played by Monsoons as in (Matthews et al. 2011) for the Lombok Strait.
  3. The role played by remote pressure forcing by Rossby modes as in (Pierini 1996) and (Panagoulia 2008).

Comment on the above.

Concluding Remarks

The manuscript should be accepted for publication after minor rewriting according to the offered suggestions.

References

Dwi Susanto, R., Leonid Mitnik, and Quanan Zheng. 2005. “Ocean Internal Waves Observed in the Lombok Strait.” Oceanography 18 (SPL.ISS. 4): 81–87. https://doi.org/10.5670/oceanog.2005.08.

Grimshaw, R., E. Pelinovsky, and T. Talipova. 1997. “The Modified Korteweg - De Vries Equation in the Theory of Large - Amplitude Internal Waves.” Nonlinear Processes in Geophysics 4 (4): 237–50. https://doi.org/10.5194/npg-4-237-1997.

Matthews, J. P., H. Aiki, S. Masuda, T. Awaji, and Y. Ishikawa. 2011. “Monsoon Regulation of Lombok Strait Internal Waves.” Journal of Geophysical Research: Oceans 116 (5). https://doi.org/10.1029/2010JC006403.

Osborne, A. R., and T. L. Burch. 1980. “Internal Solitons in the Andaman Sea.” Science 208 (4443): 451–60. https://doi.org/10.1126/science.208.4443.451.

Panagoulia, D. 2008. Mechanics of sediments (book in Greek).

Pierini, Stefano. 1996. “Topographic Rossby Modes in the Strait of Sicily.” Journal of Geophysical Research C: Oceans 101 (C3): 6429–40. https://doi.org/10.1029/95JC03138.

Syamsudin, Fadli, Naokazu Taniguchi, Chuanzheng Zhang, Aruni Dinan Hanifa, Guangming Li, Minmo Chen, Hidemi Mutsuda, et al. 2019. “Observing Internal Solitary Waves in the Lombok Strait by Coastal Acoustic Tomography.” Geophysical Research Letters 46 (17–18): 10475–83. https://doi.org/10.1029/2019GL084595.

Author Response

The authors are grateful to the reviewer for their appreciation of the work done.

Reviewer 3 Report

The manuscript entitles “Modeling of nonhydrostatic dynamics and hydrology of the Lombok Strait” written by Androsov and co-authors presents a quite interesting and advanced approach for modeling of flow in the strait connecting the Pacific and the Indian Oceans. The strait is located between Bali and Lombok islands, which are the parts of Indonesia. The applied approach includes an adaptation of the mathematical model as well as the development and implementation of the numerical algorithm. It’s an advanced and quite sophisticated approach. Undoubtedly it’s the advantage of the presented research. However, the presentation of the scientific works, results, and achievements requires some corrections. In general, the text is too long, though, some important elements are missing.

The major disadvantages of the manuscript are as follows:

  1. The Introduction is too long the composition of this part is wrong. It includes materials for a few sections. I think the paper should include a separate "Description of the case study" and "Materials". In the Introduction the Authors should focus on a general description of the problem, motivations for such studies, and the importance of their research supported with the proper literature.

The rest of the detailed information should be moved to one of the two mentioned sections. The Authors should also consider the removal of some text. The paper may be aimed at one of two: (1) detailed description and analysis of processes observed in the case study or (2) modeling of the flow phenomena in the case study. It's not possible to cover these two topics in a single paper due to the volume limitations.

If the Authors focus on the modeling, the description of the case study should only include the information necessary for understanding the process of model configuration. The rest of the text increases only the volume of this publication, but it's not important for the entire research.

  1. The purpose of the study should also be clearly defined in the Introduction. It's present in this text (lines 184-185), but the length of the Introduction causes the effects of boredom and tiredness before the reader reaches it. Definitely, the Introduction has to be shorter and the purpose has to be written earlier
  2. The nature of basic equations (1) – (6) should be discussed shortly. It should be clear what these formulae represent, e.g. mass balance, momentum balance, etc.. The particular terms in the equations could be also explained, e.g. term representing gravity force, etc.. The equations have their meaning. Such a general description may explain to the readers the character of the physical process behind the symbols and operators.
  3. In my opinion, the description of the model transformations presented in subsection 2.2 is not so important for the entire text. In general, the readers of the Water journal may not be interested in following these derivations. It should be shorter.
  4. The improve the clarity of the presented approach the description of the boundary conditions (lines 257-277) should be supported with some map, where the location of the particular conditions is marked. The conditions could be also shown in the left panel of Figure 2.
  5. Results are interesting, but they are too long once again. This section is 10-pages long. The Authors should choose what is worth to be presented.
  6. The Conclusions should not be limited to the summary only. I would expect some information about the importance of the findings, their meaning for the field of the research, etc..

There are also some smaller needs for improvement shown below

  • Line 52: I suggest using more typical symbols than ??. The paper will be published as open access. Some fonts may not be installed and displayed by default everywhere. I think the change of the applied symbols into some more typical let to avoid potential conflicts of the software after publication.
  • Line 96: The mentioned mooring stations should be marked in the map (Fig. 1)
  • Lines 98-102: These data should be listed and discussed in the section "Materials". It should be clearly explained what is their characteristics, accuracy, frequency, etc..
  • Lines 135-142: This is a rather educational passage. It could be used in the section "Methods" to explain the construction of the applied model. However, the Introduction is the proper place for more general information.
  • Lines 143-151: I understand that some of the previous passages here are used to stress the need for applied innovation, namely non-hydrostatic pressure modeling. However, this and the exactly previous passage should be in the Methods/Model section, in my opinion.
  • Lines 184-185: As I mentioned before it is the definition of the purpose, right? Don't you want to write that the implementation of the non-hydrostatic model is important for you?
  • Lines 188-203: This passage describing the content of each section is completely useless, but it makes the text longer.
  • Section Model: I suggest changing the head of this section into “Methods”.
  • Equation (1): What is q? It's not explained in the text.
  • Lines 217-218: This notation of two-component vectors, v and ∇2, is in contrary to the dimensionality of the problem. I understand why these reduced vectors are used. But it could be more consistent with the entire PDE if the vectors have three components, but the lasts are zeros.
  • Line 221: I don't understand why you need two coefficients for turbulent (eddy) diffusion. I also don't understand why the subscripts suggest their dependence on the transported constituents. As I know the turbulent mixing depends mainly on the intensity of the eddies. Could you explain the different nature of these two coefficients?
  • Lines 222-223: The eddy diffusivities seem to be described with the so-called mixing length model. If my intuition is correct, I suggest using this name.
  • Line 223: Could you prove that the Coriolis force is really important in this case? It’s a very specific component, which is neglected in some applications. Hence, it would be good to prove that this case needs the application of this term.
  • Line 231: It should be written explicitly, that this is external pressure on the water surface.
  • Line 234: What is the reference value? 1000 kg/m3?
  • Subsection 3.1: It should be a part of the Model/Methods.
  • Lines 321-322: What is the vertical resolution of these data?
  • Lines 323-324: These data should be described better.
  • Lines 337-339: This description is too short. More information has to be provided, e.g. min-max sizes, etc..
  • Line 362: A verb is missing between “the first number” and “the sea level”.
  • Lines 362-363: It's not clear what are Z0, Z1, and Z2. I've seen it's explained in the next passages, but it should be understood here.
  • Figure 5-6: It's not clear what kind of values are in the horizontal axis.

And finally, I would like to notice something, what is no formal requirement, but rather a good habit. The presented research is focused on the very specific marine area located in Indonesia. In my opinion, it would be good to invite some local researches for cooperation. In general, it could help to understand the specifications of the area, it simplifies the access to the local data and it helps in several different ways. It enables the international recognition of the researcher working in this region.

Author Response

We are grateful to the reviewer for the criticisms and hope that once corrected, the article could be published and be a starting point for the further study of this highly challenging and interesting object. A possible publication will serve for a widening of our scientific web.

Round 2

Reviewer 3 Report

  Congratulations. I see your effort made for the improvement of your text. I'm going to recommend the acceptance of your manuscript in the current form.